# Systemic Population Segmentation Based on the Unified Care Model: An Approach to Health System Transformation

**DOI:** 10.3390/healthcare13212724

**Published:** 2025-10-28

**Authors:** Yun Hu, Wah Yean Lee, Ken Wah Teo, Yeuk Fan Ng

**Affiliations:** 1Khoo Teck Puat Hospital & Yishun Community Hospital, National Healthcare Group, 90 Yishun Central, Singapore 768828, Singapore; 2Division of Population Health and Integrated Care, Singapore General Hospital, 10 Hospital Boulevard, Singapore 168582, Singapore; 3Saw Swee Hock School of Public Health, National University of Singapore, 12 Science Drive 2, Singapore 117549, Singapore; 4Health Services Research & Population Health Program, Duke-NUS Medical School, 8 College Road, Singapore 169857, Singapore

**Keywords:** systemic design, population segmentation, needs-based segmentation, health system transformation, unified care model, integrated care

## Abstract

**Context**: Population segmentation is a critical health system planning activity that enables more integrated, needs-responsive, and sustainable care. This paper describes the development and evaluation of a Systemic Health System Population Segmentation Model based on the person-centred and needs-based Unified Care Model by Yishun Health, a regional population health system in Singapore. We highlight three implications to enhance health systems operational relevance: (i) psychosocial factors as key determinants of outcomes, (ii) accountability and resource allocation across differentiated segments, and (iii) integration of lifelong and episodic care needs. **Methods**: Three interdependent models were developed, a Lifelong Care Segmentation Model, a Needs-Based Sub-Segmentation Model, and an Episodic Care Segmentation Model, all underpinned by the Unified Care Model. These models systematically stratify residents into mutually exclusive and collectively exhaustive population groups based on biopsychosocial needs across different health system levels. An expert-driven design process was used, supported by integrated administrative and clinical data. Model evaluation examined the ability to stratify patients into distinct risk groups using healthcare utilisation, costs, and readmission outcomes. **Findings**: In 2022, 78,810 residents were segmented into seven lifelong care segments, with 43,473 residents with chronic conditions further stratified into sub-segments reflecting varying complexity and psychosocial needs. Additionally, 14,335 emergency admissions were categorised into six episodic care segments. Healthcare utilisation and annual healthcare costs differed significantly across needs-based sub-segments (*p* < 0.001). Higher episodic care needs were associated with longer hospital stays, higher rates of emergency readmissions, and admission costs (*p* < 0.001). Psychosocial issues consistently emerged as a key determinant of poorer outcomes, underscoring implications for more systemic and systematic accountability assignment and more deliberate resource planning, especially for care integration horizontally. The integration of lifelong and episodic care needs further enabled operational redesign for vertically integrated health systems. **Conclusions**: By incorporating psychosocial drivers, focusing on clarifying accountability and resource allocation, and lifelong-episodic care integration, our Systemic Health System Population Segmentation Model strengthens the operational utility of segmentation as a foundation for population health system transformation and provided a robust framework for health systems governance and leadership system redesign globally.

## 1. Introduction

### 1.1. Population Segmentation in Health and Healthcare Systems

Population segmentation is a critical planning and development activity in health systems aimed at enabling more efficient system functioning. It forms a core component of value-driven strategies for improving outcomes and value in health and healthcare systems.

Segmentation uses defined criteria representing specific characteristics of the population to design “segmentation models”. The aim of segmentation models is to distribute the population into homogenous “sub-populations” with similar characteristics [1]. These defined criteria or specific characteristics are usually chosen based on their potential impact on the health systems’ outcomes-of-interest, and inability to address or intervene on these specific characteristics therefore, in effect, represents a risk of “non-attainment” of the outcomes-of-interest.

Effective segmentation models enable stratification of the population into smaller sub-populations based on different levels of risk. This stratification allows for the design of more targeted “segment-specific interventions” that can address specific characteristics more appropriately and efficiently, thereby mitigating the risks of poor outcomes [1]. Consequently, the use of segmentation models enables more targeted services planning and resource allocation, increasing overall health services and system efficiency.

Population segmentation can be implemented at the macrosystem, mesosystem, or microsystem levels of a health system, optimising different outcomes depending on the system level [1,2]. Macrosystem segmentation models are typically applied at the level of the whole population enrolled with a health system and are usually designed by the governance and leadership system of health systems. Macrosystem segmentation models are focused on forming mutually exclusive and collectively exhaustive subgroups based on differential levels of overall risk for key outcomes such as healthcare utilisation and overall healthcare costs [1]. For example, macrosystem segmentation might stratify the entire resident population into mutually exclusive groups, identifying a high-risk segment of residents living with multiple, advanced chronic diseases. This enables health system leaders to more effectively plan financing models, allocate resources, and set accountability structures for this segment, as for all segments, across the population health system.

In contrast, mesosystem and microsystem segmentation models focus on specific sub-populations within the health system. These models aim to improve outcomes relevant to specific providers, programmes, or organisations, often for specialised programme redesign [3,4]. Additionally, microsystem segmentation models may employ disease-, discipline-, and/or profession-based clinical or specialty criteria to generate patient segments with different disease/sub-specialty severity, to facilitate the design of disease- or site-based care paths for provider teams. The primary goals are to optimise condition-specific outcomes and efficiently allocate specialised resources. In relation to the aforementioned high-risk segment of residents living with multiple, advanced chronic diseases, depending on the residents’ unique disease conditions or the provider’s services criteria, each resident would belong in multiple mesosystem or microsystem segmentation model segments.

It is important to note that while mesosystem and microsystem segmentation models serve a valuable purpose, they often have considerable overlaps and gaps in their specific characteristics and outcomes of interest. Unlike macrosystem segmentation models, they are usually neither mutually exclusive nor collectively exhaustive of the whole population.

### 1.2. International Context

With population ageing and increasing chronic disease burden [5], health systems worldwide are experiencing escalating costs and facing challenges in ensuring sustainable access to quality healthcare services. As a result, health system financing mechanisms are increasingly moving away from “fee-for-service” towards “bundled funding”, or other population-based funding mechanisms such as “capitation”, to sustain patient care outcomes within more constrained financial resources [6,7]. These financing policy changes effectively pass the financial risks of healthcare provision, from the level of whole health systems down to subsidiary systems or organisations, such as provider systems (e.g., integrated care organisations or hospital systems) or third-party organisations (e.g., accountable care organisations or health insurers).

This shift in financial risk has significant implications for population health systems’ planning and delivery of healthcare. As subsidiary provider systems and healthcare organisations take on this financial risk, they must consider macrosystem healthcare utilisation and overall cost-of-care as key outcomes that their mesosystem and microsystem levels ultimately contribute towards. This shift necessitates a rethinking and redesign of segmentation models commonly used by these entities for mesosystem and microsystem services and resource planning and allocation, especially in relation to how they work together systemically to enable macrosystem populations’ segment outcomes more effectively.

### 1.3. Local Context

In response to these changes, we have observed a trend in Singapore where hospital systems are exploring the use of macrosystem population segmentation models for patients within their geographical catchment areas. The primary aim is to reduce the utilisation of expensive hospital care and, consequently, costs-of-care [8]. The expectation to manage financial risks has also driven Singapore hospital systems to initiate “upstream” macrosystem population segment-specific interventions to reduce healthcare costs and mitigate financial risks [9].

In 2017, the National Healthcare Group (NHG), reorganised Khoo Teck Puat Hospital (a 795-bed acute hospital), Yishun Community Hospital (a 428-bed sub-acute, rehabilitative, and palliative care hospital), and Admiralty Medical Centre (an ambulatory care centre) into Yishun Health (YH) as an integrated care organisation serving 320,000 residents within the Yishun Zone Regional Population Health System in Northern Singapore. YH’s role expanded beyond mere healthcare service provision to collaborating with care partners to enhance residents’ health and well-being and integrate care [10]. Plans were also developed for Yishun Health to manage age-specific healthcare utilisation, cost-of-care outcomes, and value, to assume a significant portion of the financial risk of healthcare services provision for residents living within the Yishun Zone.

Operating within this context, YH initiated numerous health system and services transformation initiatives to accelerate the development of a people-centred, integrated, and value-driven regional population health system. A key initiative was the development and adoption of a set of design principles for the Unified Care Model (UCM) with its subsidiary lifelong care and episodic care services models to guide the development of the YH future-state integrated and value-based Service Delivery System.

The UCM incorporates several key features that influence population segmentation. First, it aims to meet population health needs, measured by resident health-related experience, quality-of-life, and protective health factors. Second, while individual provider outcomes are important, they are considered subsidiary components of broader population health outcomes. Third, the system measures costs across all providers to optimise resident and patient value. Fourth, integrated care models—lifelong care and episodic care—address the totality of health needs within the Yishun Zone population. The lifelong care subsystem focuses on ongoing health maintenance and chronic disease management, while the episodic care subsystem addresses acute crises and complex elective medical events. These subsystems collectively cover all residents with health needs in the Yishun Zone. Finally, both systems have microsystems representing smaller sub-populations with distinct needs, such as people living well and those with advanced chronic illness in lifelong care and patients requiring acute stroke and pneumonia care in episodic care.

### 1.4. Care Model Implications for Health System Population Segmentation Models

Population segmentation models are increasingly utilised to optimise outcomes across various levels of the health system [11]. However, the rationale behind selecting specific criteria for population segmentation are often not immediately apparent or transparent. It is also unclear how the different population segment-specific interventions that help to achieve outcomes for their respective population segments, whether and how they are vertically integrated with segment-specific interventions at the macrosystem level, and systemically, whether they come together effectively and efficiently in generating whole-health-system outcomes.

For instance, instead of focusing on characteristics directly impacting a person’s lifelong functional health outcomes in spite of their acute disease condition, a hospital provider system may deploy microsystem and mesosystem segmentation models based upon provider-centric criteria such as medical subspecialty or disease-specific criteria only, and then try to improve patient function and experience outcomes only within the context (and competencies) of a medical specialty or disease treatment pathway. In other words, coordination of continuing care to attain person-centred functional outcomes is beyond the ambit of particular medical subspecialty or disease-specific pathways and becomes left to other, often less resourced, care providers elsewhere in the health system (macrosystem).

The result, therefore, is that there can be different degrees of disconnection between a health system’s stated goals of person-centred outcomes and the actual criteria used for population segmentation at the microsystem, mesosystem, and macrosystem levels. Indeed, the causal relationship between selected characteristics for population segmentation and the health system’s desired outcomes can be complex and, in some instances, insufficiently examined and tenuous, or even wrong. This does not mean that desired person-centred health system outcomes are unattainable. However, segmentation-specific interventions based on population segmentation models that are not needs-based or systemically linked to macrosystem segmentation models are unlikely to be effectively designed and even less likely to be efficient in improving person-centred whole-health-system outcomes and how this can be achieved is a key research gap in the literature. While existing segmentation models at the mesosystem and microsystem levels remain relevant, especially for subsidiary systems and provider organisations, there is an increasing need to ensure that mesosystem and microsystem outcomes align with the desired outcomes at the macrosystem level given the context of broader prevention strategies and sustainable health system cost management. New approaches to population segmentation at the mesosystem and microsystem levels must address these tensions while maintaining operational efficiency and avoiding excessive cost-cutting or care rationing. This situation therefore demands a comprehensive and systemic design approach to population segmentation.

### 1.5. Systemic Health System Population Segmentation—Singapore Case Study

In a well-defined health system, changes to the context and therefore the definition of the health systems’ macrosystem outcomes and its macrosystem population segmentation model will have cascading effects. The criteria for mesosystem (e.g., cohorts) and microsystem (e.g., high-risk populations) segmentation models and their interdependencies with each other within the broader macrosystem framework will each require re-definition or clarification to ensure their distinctiveness and efficiency across different levels of the health system.

The changing local context of YH has therefore driven us to embark on systemic redesign and comprehensive redefinition of our population segmentation criteria and models spanning the macrosystem, mesosystem, and microsystem levels of our regional population health system’s service delivery model based on the UCM.

We adopted population needs-based criteria [8] to understand health determinants across the entire population to facilitate more effective delivery of person-centred health services to sub-populations or segments with similar health needs, regardless of individual disease status [12,13]. Recognising the continued importance of mesosystem and microsystem segmentation models within a macrosystem population health system, we therefore also propose a clear relationship and interdependency between population needs-based segmentation models at the macrosystem level and programme-criteria based (utilisation-based) or disease-treatment based models or at the mesosystem and microsystem levels. This paper aims to

Introduce a novel person-centred and needs-based Systemic Health System Population Segmentation Model approach to enable whole population health system redesign.Describe the Lifelong Care Segmentation and the Needs-Based Sub-Segmentation Models for the macrosystem and mesosystem levels of the Service Delivery System of a health system, based on the UCM.Describe the Episodic Care Segmentation Model for the mesosystem and microsystem levels of the Service Delivery System of a health system, based on the UCM.Illustrate the operational relevance of such a systemic segmentation model for integrated care delivery by highlighting the implications of psychosocial determinants of health outcomes and the accountability and resource allocation considerations at different system levels, as well as the vertical integration of lifelong and episodic care within health systems.Illustrate the evaluation results of these models.

These clarifications will clarify how various population segmentation models can work together to provide a systemic approach to health system design, enabling more systemic services planning and resource allocation to enhance care integration, outcomes, and value at the whole population health system level. Such an approach will facilitate more effective segment-specific integrated care delivery and service planning [14] and provide insights that may be valuable to other health systems facing similar challenges in population health management and integrated care delivery.

## 2. Materials and Methods

The National Healthcare Group (NHG) had developed a River of Life (ROL) framework to meet health needs for the entire population segmented to five segments of care covering both lifelong care needs—living well, living with illness, living with frailty, and leaving well—and episodic care needs—crisis and complex care [15]. However, biological conditions, e.g., presence of chronic disease and/or frailty status alone, do not fully reflect the lifelong needs of the population [16]. Also, the disease and/or frail groups could have very diverse needs, which warrants further sub-segmentation to enable more targeted interventions. Furthermore, episodic care needs are often triggered in the form of an escalation, rather than in isolation, from the base lifelong needs. Yishun Health (YH) enhanced the ROL segmentation model by 1. incorporating the stage of disease trajectory and population psychosocial needs into the lifelong segmentation (LS) model; 2. further sub-segmenting population with chronic diseases and/or frailty into the Needs-Based Sub-Segmentation (NBSS) model; and 3. incorporating sporadic needs into lifelong base needs to develop an Episodic Segmentation (ES) model [17].

There are two major approaches to population segmentation tool development: an expert-driven approach and a data-driven approach [8]. YH adopted an expert-driven approach in the interim to ensure segmentation criteria aligned with the Unified Care Model (UCM)’s person-centred philosophy and incorporation of established clinical knowledge about disease progression and psychosocial complexity. Internally, this expert-driven approach facilitated greater acceptance amongst our clinicians to use these segments for services planning while initiating the collection of more comprehensive population needs and outcomes data through our Yishun Health Population Health Survey 2022. A multidisciplinary team of internal experts was assembled, comprising clinicians, public health and health system and services planning consultants, and data scientists and data analysts. This diverse group collaboratively selected and defined both the population inclusion and exclusion criteria and the segmentation criteria, ensuring a holistic perspective in the development process.

### 2.1. Lifelong Care Segmentation (LS) Model Development

The development of the LS model was underpinned by the following key guiding principles: (1) Instead of using a provider-centred approach such as disease condition- or healthcare utilisation-based criteria, we adopted a set of person-centred and needs-based segmentation criteria. This means that the model must consider the entire population and the full spectrum of needs that influences an individual’s health outcomes. Based on a literature review [18,19,20,21,22,23] and inputs from our team of experts, we selected segmentation criteria that consider an individual’s holistic needs across their bio-psycho-social spectrum. These criteria encompass the stage of chronic illness, the presence of mental health conditions, and social issues [17]. (2) Segments must be sufficiently discriminatory and mutually exclusive, ensuring that the population within each segment shares similar health needs. Conversely, across each segment, people should have systematically well-differentiated healthcare needs. (3) Additionally, segments must collectively be exhaustive of all residents in Yishun Zone. By adhering to these criteria, the LS aims to assist policy makers in planning services, evaluating outcomes, and tracking progress.

We first identified the total number of residents living in the Yishun Zone Planning Area based on census data from the Singapore Department of Statistics (2022) [18], including Singapore citizens or permanent residents who reside in the catchment area of Yishun Zone. Next, using the existing and readily available data points that reflect or serve as surrogates for the segmentation criteria mentioned above, we identified residents who visited YH in 2022 for lifelong segmentation. These data points were found within YH’s Central Data Repository (CDR), a comprehensive data repository that integrates administrative, clinical, and financial data from multiple IT source systems within the organisation. A list of 38 chronic illnesses and their corresponding ICD-10 codes was used to identify if a resident had prior chronic disease diagnoses (Appendix A). These chronic conditions were selected based on the NHG’s established chronic disease list, with additional input from YH clinicians to ensure relevance to the local population’s disease burden and care delivery priorities. All past historical primary and secondary ICD-10 diagnosis codes of the residents were included in the analysis. To ascertain if the chronic illnesses were “early” or “advanced” in terms of disease progression, we referred to the resident’s Diagnosis-Related Groups (DRGs), ICD-10 codes, and SNOMED codes of their chronic disease diagnosis. If the chronic disease diagnosis codes indicated the presence of complications or complications were identified by clinician’s review, we defined such residents to be at an advance stage of chronic illness. In addition, patients with Clinical Frailty Score (CFS) statuses with a cut-off score >7 were classified under the “Leaving Well” segment.

Next, the evidence of psychosocial complexity impacting healthcare needs and outcomes necessitated the operationalisation of relevant psychosocial needs criteria into our models. As there were significant administrative data limitations within our existing systems, we used available proxy indicators within our CDR to initiate the process i.e., “just get started”. We defined psychosocial issues as the presence of mental health conditions, social issues, or both. Mental health issues were identified through ICD-10 diagnostic codes for mental and behavioural disorders (F00–99 categories) from residents’ historical primary and secondary diagnoses. This included conditions such as depression, anxiety disorders, bipolar disorder, and schizophrenia documented during clinical encounters. Social issues were identified through two proxy indicators: documented visits to medical social workers, indicating psychosocial needs requiring professional intervention, and residence in public rental housing as a proxy for low socioeconomic status, since such housing is allocated to very-low-income households in Singapore. After applying the above segmentation criteria, seven lifelong care segments with increasing complexity of lifelong care needs were derived forming our needs-based Lifelong Care Segmentation Model. The seven lifelong care segments (LSs) are: LS1. Living Well, LS2. Living Well with Social Issues, LS3. Early Disease, LS4. Early Disease with Psychosocial Issues, LS5. Advanced Disease, LS6. Advanced Disease with Psychosocial Issues, and LS7. Leaving Well.

### 2.2. Further Needs-Based Sub-Segmentation (NBSS) Model Development

To facilitate the design of more targeted, sub-segment-specific services, for segments with more intense needs, the NBSS model further stratifies residents with chronic diseases (LS3–LS6) into ten sub-segments based on disease complexity and psychosocial needs. The NBSS model creates a matrix structure with two main dimensions: disease stage (early versus advanced) and complexity level. Sub-segments range from residents with single chronic diseases only (A1, B1) to those with multiple chronic diseases combined with both mental health issues and social problems (A5, B5). The intermediate categories capture residents with multiple chronic diseases alone (A2, B2), multiple chronic diseases with mental health issues (A3, B3), and multiple chronic diseases with social issues (A4, B4). This granular sub-segmentation aims to enable the development of horizontally integrated services models that facilitated more targeted care planning for residents with differing levels of medical and psychosocial complexity within the broader chronic disease population. (Detailed segmentation criteria are provided in Appendix B.)

### 2.3. Episodic Care Segmentation (ES) Model Development

Whilst lifelong care segments represent the ongoing, base needs of a resident living in the community, when crisis and complex care needs occur, there is an escalation of care needs. Under the UCM, this is termed episodic care needs and leads to, e.g., urgent acute hospital admissions. Therefore, the ES model was developed as a systemic extension of the LS model specifically for residents admitted to YH through the emergency department. This model incorporates different levels of episodic care needs, as quantified by the Patient Acuity Category Scale (PACS) value. By layering acute care needs over the existing LS model, the ES model provides a more comprehensive yet systemic view of differing patient person-centred needs during their acute episodes, facilitating vertically integrated and more systemic care delivery for the whole health system. With reference to average length of stay (ALOS) as the key outcome, six ES segments were derived based on Lifelong Segment and ED PACS Value, namely, ES1. Low Resource Intensity, ES2. Low Acuity, ES3. Low Acuity with Psychosocial Issues, ES4. High Acuity, ES5. High Acuity with Psychosocial Issues, and ES6. Leaving Well (Figure 1).

### 2.4. Health System Population Segmentation Model Evaluation

Operationally, our Systemic Health System Population Segmentation Model is meant to allow health system stewards to create additional accountability structures by segments that highlight the disproportionate effect of psychosocial factors on utilisation and outcomes, assigning responsibility for outcomes to providers across the care continuum systemically. Accordingly, evaluation was aimed at understanding the respective segmentation model’s ability to segment Yishun Zone residents into mutually exclusive groups that are relatively homogenous in terms of lifelong and episodic healthcare needs, and that these groups were meaningfully distinct—crucial in rationalising the customisation of care plans and the dedication of resources to each segment.

To evaluate whether our NBSS model was able to generate segments of people with uniquely distinct needs, we correlated the segments to various population health outcomes [19,20,21,22,23,24] or surrogates, starting with readily available data in our CDR, including healthcare utilisation and cost in the latest year that residents visited YH, i.e., number and cost of visits and admissions to our hospital’s emergency department, specialist outpatient clinics, and inpatient ward, and the annual hospitalisation bed days. The statistical distinction and difference between population segments across the outcome measures was also studied.

To evaluate the ability of our ES model to generate groups of patients with distinct level of episodic care needs, statistical differences between the ES segments were examined using available outcome data, i.e., ALOS, inpatient admission cost, and 30-day emergency readmission.

Administrative data from the electronic medical record system were used, with minimal missing values due to systematic data collection. Residents with incomplete essential segmentation data were excluded. Data preparation was conducted using the Konstanz Information Miner (KNIME) Analytics Platform version 4.6.3 (Switzerland). Statistical analyses were performed with SPSS version 27. Chi-square tests were used for categorical variables to assess associations between segments and demographic characteristics. One-way ANOVA was applied to compare continuous outcome variables (e.g., utilisation, costs, length of stay) simultaneously across multiple segments while controlling for error. Linear regression was used to examine associations between segmentation variables and continuous outcomes, adjusting for potential confounders. Parametric tests were selected given the large sample size and approximate normality of the outcome distributions. Statistical significance was set at *p* < 0.05. AI software was used for proofreading (Pair Chat, Open Government Products, Singapore; ChatGPT (GPT-5), OpenAI, USA).

## 3. Results

In the year 2022, a total of 78,810 residents known to Yishun Health were segmented into seven lifelong care segments using our Lifelong Care (LS) Segmentation Model (Figure 2).

The majority of residents were classified as Living Well (43.1%) or Living with Early Disease (33.0%), representing relatively healthy populations without significant psychosocial complexity. As expected, age and comorbidity burden increased progressively across lifelong care segments (LS1–LS7), with mean age rising from 38.9 to 76.9 years and Charlson Comorbidity Index scores increasing from 0.4 to 8.8, reflecting the clinical logic of the segmentation model (Table 1). The CCI is the most widely used index used to determine survival rates in patients with multiple comorbidities [25].

The Needs-Based Sub-Segmentation (NBSS) Model further sub-segmented 43,473 residents with chronic disease (Figure 3). Most residents were at early disease stages without psychosocial needs (A1 and A2), while fewer had mental and/or social issues (A3–A5 and B3–B5). Residents with complex needs were older with higher Charlson Comorbidity Index scores (Table 2), following expected clinical patterns.

Table 3 demonstrates that the NBSS model effectively predicts healthcare outcomes, with sub-segments showing progressively higher utilisation and costs corresponding to increased complexity. Residents with multiple chronic diseases, mental health issues, and/or social problems had significantly higher emergency department visits, specialist consultations, inpatient admissions, hospitalised bed days, and annual healthcare costs. Notably, mental health and social issues emerged as stronger predictors of poor outcomes than did medical complexity alone.

Next, our Episodic Care Segmentation (ES) Model classified 14,335 emergency admissions into six segments (ES1–ES6), with most categorised as low acuity with or without psychosocial issues (ES1–ES3). Higher-acuity segments and those with mental/social issues demonstrated significantly longer length of stay, higher costs, and increased 30-day readmission rates (Table 4). Notably, psychosocial issues were the primary driver of poor outcomes even among patients with similar medical acuity levels.

A critical finding across all segmentation models was the profound impact of psychosocial issues on healthcare outcomes, which consistently emerged as a stronger predictor of poor outcomes than medical complexity alone. In the NBSS model, residents with psychosocial issues (mental health and/or social problems) demonstrated significantly higher healthcare utilisation and costs compared to those with similar medical complexity but without psychosocial issues. For example, residents with early disease and psychosocial issues (A3–A5) had higher emergency department visits (0.9–1.6 vs. 0.4–0.6), longer hospitalisation stays (2.3–9.5 vs. 0.7–0.8 days), and substantially higher annual costs (SGD 3981–SGD 10,682 vs. SGD 1941–SGD 2538) than those with early disease alone (A1–A2).

Similarly, in the ES model, the presence of psychosocial issues was the key differentiator of outcomes within similar acuity levels. Patients with low acuity but psychosocial issues (ES3) had dramatically higher 30-day readmission rates (16.3% vs. 5.3%) and longer stays (8.2 vs. 4.3 days) compared to those with low acuity alone (ES2). This pattern persisted across all acuity levels, with psychosocial issues consistently associated with poorer outcomes regardless of medical complexity.

These results emphasise that psychosocial factors, more than medical complexity alone, drive poor outcomes. This has immediate implications for health system planners to redirect resources towards social and mental health support within community and hospital care.

## 4. Discussion

Population segmentation models, operating at different levels of a health system, are designed for various legitimate purposes but lack systemic focus and coordination across system levels to achieve a common, integrated whole-health-system goal. In essence, macrosystem segmentation models can be thought of as “demand-sided” segmentation models representing the needs of residents and patients that can be used to align or harness the care by different “supply-sided” segmentation models and their integrated services models focusing on meeting the needs of specific cohorts of interest to a profession/specialty, a disease type, or an institution and the services they provide, e.g., acute care services.

Supply-sided mesosystem- or microsystem-level patient segmentation models can generate “mutually exclusive sub-segments” that exhaustively include all patients meeting the service model’s inclusion criteria. These models may individually lead to more patient-centric care for specific disease-based patient populations. However, from a systemic perspective, when taken together at the level of the macrosystem or population health system, these numerous supply-sided mesosystem and microsystem-level segmentation models cease to be mutually exclusive. Since patients are cared for by providers and processes designed for multiple overlapping patient segments, supply-sided patient segmentation models paradoxically undermine the patient-centricity they aim to create. Ultimately, this undermines the ability to achieve genuine integrated and value-driven care. [26]. In addition, disease- and healthcare utilisation-based criteria appear to have been used by provider-oriented systems to design segmentation models that are actually meant for their macrosystem or their population health systems [8,14,27]. We posit that this is a fundamental misunderstanding of the criteria driving the type and quantity of risks to outcome attainment at the macrosystem level versus at lower levels of the system, such as at the mesosystem and microsystem levels of a population health system.

Clarifying these distinctions—as we did using a systemic cascade of models in our whole-health-system population segmentation model—is essential and will be beneficial for improving the person centricity and the value of population health systems.

### 4.1. Psychosocial Factors as Determinants of More Systemic Outcomes

First, our findings mirror a growing body of international literature highlighting that psychosocial determinants, including mental health conditions, social isolation, and socioeconomic disadvantage, often exert a greater influence on healthcare utilisation and outcomes than medical complexity alone [28,29,30]. Macrosystems should therefore adopt psychosocial factors as key determinants for more person-centred outcomes, moving beyond a narrow focus on cost and utilisation as segmentation criteria. By prioritising these factors, health system stewards can also better align with individual and community needs and meet rising public expectations. Mesosystems and microsystems, which have traditionally focused on specialty or disease-based or utilisation-based criteria for segmentation so as to improve the same outcomes, can in turn incorporate psychosocial factors into their segmentation criteria to align services design towards more person-centric and value-based outcomes that add up more systemically to whole-system, macrosystem outcomes. Such services development can be guided using person-centred, whole system models of care such as the Unified Care Model (UCM) [10], to ensure effectiveness at all system levels.

### 4.2. Accountability and Resource Allocation for All Population Segments

Second, once we have a systemic cascade of person-centred and needs-based population segmentation models, i.e., Lifelong Care Segmentation (LS), Needs-Based Sub-Segmentation (NBSS), and Episodic Care Segmentation (ES), the purpose of care improvement, integration, and health system transformation—which is thought of as the pursuit of services redesign and resource allocation that more effectively address these specific characteristics to enhance outcomes or mitigate the risks of poor outcomes for different population segments or sub-populations [11,31]—can now be pursued systemically, at scale, potentially enabling more efficient whole-system allocation of resources. Health systems and services design can prioritise policy to encourage primary healthcare providers to be more accountable to manage the health and medical conditions of these residents over a specified duration, e.g., annual or longer “cycles” of care and therefore resident health outcomes. For example, we can use the NBSS model to assign residents with similar needs to respective Primary Coordinating Doctors (PCDs) and their care teams to be accountable for their lifelong health outcomes [32]. This model enables PCDs to estimate workload, plan resources, and design horizontally integrated segment-specific interventions to meet residents’ needs.

Indeed, after applying the NBSS model, we found that most residents have lower health needs, with no or early-stage diseases. PCDs can emphasise primary prevention and routine chronic disease management as necessary. A smaller number of residents have multiple or complex medical conditions that involve mental and/or social issues driving significantly higher healthcare utilisation and costs. For this segment, it may be possible to commission PCDs to lead a multi-disciplinary, multi-institution team of more skilled professionals to be accountable for integrated services models that incorporate more personalised care and support processes that address mental, social and medical risk factors together to improve patient outcomes.

### 4.3. Vertical Integration of Lifelong and Episodic Care

Third, beyond horizontally integrated care teams, the Systemic Health System Population Segmentation Model can also provide guidance for vertical integration of services models across different health system levels that together enable more coherent whole-health-system planning to meet residents’ needs and improve population health outcomes.

As explained, LS and NBSS models operationalised by PCDs and their teams aim to bring about care that is more horizontally integrated. However, the ES model is also derived from the LS and NBSS models and therefore advocates for the bringing of lifelong person-centred perspectives into episodic care to improve episodic care design and outcomes. For acute hospitals, this offers an innovative approach to patient care and resource allocation, e.g., operational bed/ward management. By segmenting patients upon admission, patients from different NBSS segments can be immediately assigned to the appropriate ES care team with the necessary bed/ward resources, such as generalists, specialists, or medical social workers, to treat and manage different levels of acuity and social issues. Service managers can consider coordinating this care with the patients’ PCDs. Such an approach can promote new and vertically integrated services models to come into being that consider the patients’ sporadic crisis care needs during their acute hospital stay alongside their lifelong needs so as to improve the resident experience of acute care while shortening unnecessary patient stays in acute hospitals due to the patients’ psychosocial determinants of health, thereby improving operational efficiency. Transitions of care back to the PCD within the community are also more continuous and coordinated, potentially lowering financial risks for both the hospital system and patients ultimately.

### 4.4. Comparison with International Models, Equity and Ethical Considerations

Our approach builds upon and goes beyond other validated or well-established international segmentation models [4]. For example, the Bridges to Health framework similarly proposed eight population groups based on health states (e.g., healthy, maternal, frail elderly, end-of-life), providing conceptual clarity for service design [1] but with less emphasis on linking segmentation to operational accountability structures across the health system. The Kaiser Permanente Risk Pyramid segments populations primarily by clinical risk of utilisation to assign patients into tiers for improved chronic disease management [33] and is therefore more operationally anchored. While the British Columbia Health System Matrix [34] expands this to the whole population, neither of these sufficiently integrates psychosocial or equity-related determinants and none aim to provide systemic guidance for integrated care operationalisation. In contrast, our Systemic Health System Population Segmentation Model—anchored using the UCM—explicitly incorporates psychosocial and socioeconomic factors; by integrating lifelong and episodic needs both horizontally and vertically, our approach extends beyond descriptive segmentation to provide a pragmatic governance framework for accountability and resource allocation for whole-health-system integrated care service redesign.

Population segmentation raises equity and ethical concerns, especially regarding resource allocation and potential stigmatisation. By incorporating socioeconomic proxies like public rental housing status, our model potentially identifies residents at higher risk of poor outcomes due to social disadvantage. These equity-sensitive criteria, when brought into services design in the form of NBSS segments, can potentially help health systems improve health equity. Last, our segmentation categories are meant to be system and services planning tools; this lessens the chance of potential patient stigma, but we acknowledge that adequate clinical and operational governance processes must ensure transparent ethical trade-offs during care to protect individual residents/patients. Future iterations of our systemic segmentation models can include richer self-reported measures of health literacy, functional status, and social support to enable more precise needs and other mental health and social criteria, including inequity identification.

## 5. Limitations and Generalisability

### 5.1. Operational Feasibility and Implementation Challenges

Front-line clinician involvement was integral to our design and refinement of the segmentation models in our lived experience. Multidisciplinary clinicians participated in defining segmentation criteria and psychosocial indicators, and their feedback through iterative patient case discussions exemplified how we manage the criteria and operationalisation of the cascade of systemic population segmentation models. Specific services development forums shaped ongoing services design and refinements to aid implementation. While many clinicians welcomed the clarity segmentation brought to patient outcomes accountability and coordination, concerns arose about workload as we transition from disease-based to bio-psycho-socially integrated segmentation approaches. For example, the LS and NBSS models faced manpower constraints in specialist care and agreement issues in primary care, while the ES model was challenged by incomplete data at the emergency department. These required significant continuous service model and organisation adaptations, which are still ongoing.

Cultural change was critical—we attempted to move clinicians and managers from viewing population systemic segmentation as an alternative to disease-based models to embracing it as a complementary, system-enabling ‘AND’ approach. Overcoming these challenges required iterative dialogue, alignment with changing institutional priorities, and demonstration of tangible value through service evaluation, and many of these efforts and organisational adaptations remain ongoing. Our experience suggests that success strongly depends on adequate resourcing, data readiness, and cultural commitment to achieving person-centred, integrated care as one whole system.

### 5.2. Outcome Attribution

Establishing systemic accountability is essential to ensure that improvements in outcomes are meaningfully linked to segmentation-enabled strategies. Yet, a persistent challenge lies in delineating accountability across multiple sites and care teams that jointly deliver vertically and horizontally integrated care and in measuring it efficiently. Our current approach anchors accountability to defined population segments—such as assigning higher-need sub-segments to specific Primary Coordinating Doctors (PCDs) and care teams—thereby allowing LS/NBSS derived segment-level measurement and tracking of utilisation, costs, and outcomes. In addition, we have also developed ES derived segment-level measurements and dashboards. However, as integration deepens, clarification of shared accountability across teams and settings will be complex. A promising direction may involve developing multi-tiered accountability frameworks that align shared outcomes with distributed responsibilities both horizontal and vertically, supported by interoperable data systems and common health systems and subsystem metrics. Ultimately, advancing this work will require future research into governance structures and measurement frameworks that can operationalise shared accountability within systemically segmented and systemically designed health systems.

### 5.3. Data and Model Validation

Our models rely predominantly on administrative data from our electronic medical record system, which excludes important psychosocial and functional measurements such as health literacy, detailed health behaviours, functional status, caregiver availability, and comprehensive social determinants of health. To address these data constraints, we incorporated available proxy indicators including medical social worker visits and public rental housing status as socioeconomic markers, utilised Clinical Frailty Scale scores where available, and engaged multidisciplinary clinical experts to ensure appropriate interpretation of administrative data. We acknowledge that we may still have underestimated the true prevalence of biopsychosocial needs, as our data only captures formally diagnosed conditions and documented social needs, missing undiagnosed or undocumented psychosocial issues. Moving forward, Yishun Health has conducted a Population Health Survey in 2022 to systematically collect resident-reported functional needs, social circumstances, and quality of life measures, which will be linked with administrative data and data from our electronic medical record system to enhance segmentation accuracy and comprehensiveness.

In addition, while our expert-driven approach to model development is clinically rational and contextually appropriate, the lack of external validation or comparison with data-driven models (e.g., clustering algorithms, latent class analysis) may limit the robustness and generalisability of our segmentation framework. To mitigate this concern, we conducted comprehensive internal validation using healthcare utilisation, cost, and readmission outcomes to demonstrate the discriminatory ability of our segmentation models across different population groups. For future enhancement, we plan to compare our expert-driven approach with data-driven segmentation methods such as machine learning clustering algorithms and latent class analysis to assess model robustness and identify potential improvements. Additionally, we are developing partnerships with other health systems to enable external validation of our segmentation framework.

### 5.4. External Validity, and Next Steps for Adoption

Notwithstanding the above challenges and our segmentation innovation being within Singapore’s urban and regional population health system context, our approach to the segmentation using the Systemic Health System Population Segmentation Model can be adapted to other health systems. Its design principles—needs-based, psychosocially informed, and integrating lifelong with episodic perspectives—can be applied to populations with differing demographics and disease burdens. In rural settings, proxies may shift toward geographic access or service availability; in younger populations, toward behavioural and preventive factors; and in systems with differing morbidity profiles, toward locally salient conditions.

Successful implementation elsewhere will require significantly different enabling conditions than those in Singapore that we have discussed above. However, the need for interoperable data infrastructure linking clinical and social domains, clear governance frameworks assigning outcome accountability to care teams, investment in multidisciplinary workforce capability, and staged piloting to iteratively refine the models are clearly areas that remain similar and needed by every health system.

Our Systemic Health System Population Segmentation Model approach not only advances academic understanding but also provides health system leaders with pragmatic levers to redesign services, allocate resources, and strengthen accountability in pursuit of integrated, person-centred care. Embedding segmentation into routine strategic planning and resource allocation cycles will help all health systems to sustain system-level impact. Future work should focus on evaluating how such systemic population segmentation models perform across diverse governance and financing contexts.

## 6. Conclusions

Yishun Health’s Systemic Health System Population Segmentation Model successfully stratified residents into distinct groups with predictable health outcomes and resource needs. The models’ key insight—that psychosocial factors drive healthcare outcomes more than medical complexity alone—has profound implications for health system design and resource prioritisation.

Beyond methodological innovation, the models’ operational value lies in clarifying the impact of psychosocial drivers, strengthening accountability and resource allocation frameworks, and integrating lifelong with episodic care needs. These models function as structural mechanisms for health system transformation, enabling targeted resource allocation, clear accountability frameworks, and coordinated, integrated care delivery across the care continuum and the whole health system. By embedding these segmentation approaches within value-based care frameworks, health systems can achieve more effective population health management and improved outcomes.

Our systematic and systemic approach offers a replicable framework for other health systems seeking to transition from fragmented, provider-centred care to integrated, person-centred population health models. For adoption beyond our setting, health systems should prioritise integrated data systems, governance alignment, workforce readiness, and phased implementation as critical next steps.

## Figures and Tables

**Figure 1 healthcare-13-02724-f001:**
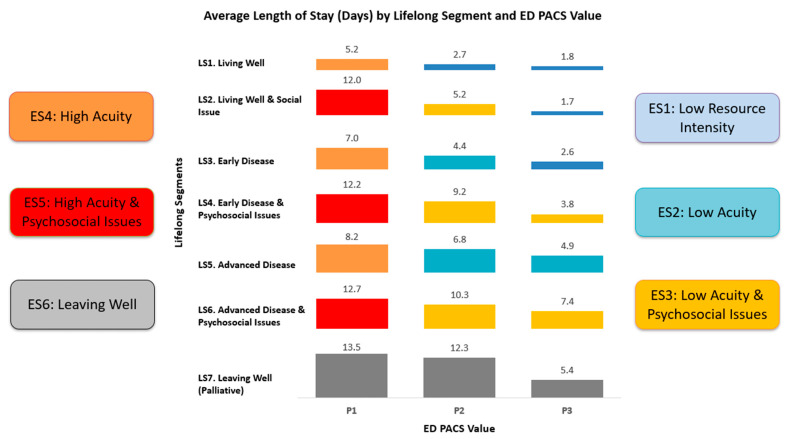
Yishun Health Episodic Care Segmentation (ES) Model and criteria.

**Figure 2 healthcare-13-02724-f002:**
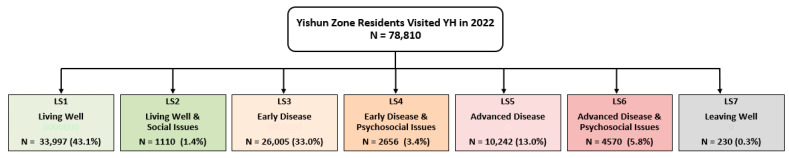
Population size in Yishun Health Lifelong Care (LS) Segmentation Model.

**Figure 3 healthcare-13-02724-f003:**
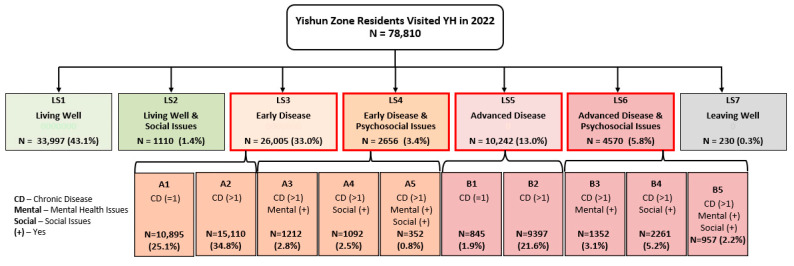
Population size in Yishun Health Needs-Based Sub-Segmentation (NBSS) Model.

**Table 1 healthcare-13-02724-t001:** Profile of Yishun Zone residents by lifelong care segments.

	LS1	LS2	LS3	LS4	LS5	LS6	LS7	*p*-Value
**Mean Age** (Years)	38.9	37.2	56.7	61.4	63.7	67.6	76.9	<0.001
**Gender**								
Female %	49.3%	49.9%	50.3%	56.9%	44.6%	49.6%	46.7%	<0.001
**Ethnicity**								
Chinese, %	59.9%	30.3%	68.8%	65.9%	62.2%	60.6%	69.6%	<0.001
Malay, %	11.5%	39.0%	11.9%	16.3%	16.5%	18.8%	13.1%	
Indian, %	11.9%	14.5%	10.5%	10.7%	13.3%	12.7%	7.0%	
Others, %	16.7%	16.2%	8.8%	7.1%	8.0%	7.9%	10.3%	
**Average CCI score**	0.4	0.4	2.2	3.1	5.1	6.4	8.8	<0.001

**Table 2 healthcare-13-02724-t002:** Profile of Yishun Zone residents by needs-based sub-segments.

	A1	A2	A3	A4	A5	B1	B2	B3	B4	B5	*p*-Value
**Mean Age** (Years)	49.3	62.1	58.2	63.5	65.4	47.0	65.2	68.6	65.2	71.6	<0.001
**Gender**											
Female %	52.6%	48.4%	57.1%	51.4%	65.3%	48.3%	43.9%	57.9%	42.5%	54.4%	<0.001
**Ethnicity**											
Chinese, %	65.5%	71.9%	72.4%	58.5%	67.3%	56.6%	62.9%	68.7%	53.6%	64.6%	<0.001
Malay, %	13.0%	11.0%	10.5%	22.1%	13.1%	17.5%	16.0%	11.1%	24.8%	15.4%	
Indian, %	11.0%	10.0%	9.8%	11.8%	12.5%	13.4%	13.7%	13.6%	12.7%	14.2%	
Others, %	10.5%	7.0%	7.3%	7.6%	7.1%	12.5%	7.3%	6.6%	8.9%	5.9%	
**Average CCI score**	1.2	2.9	2.7	3.3	3.7	1.8	5.4	6.1	6.2	7.3	<0.001

**Table 3 healthcare-13-02724-t003:** Healthcare utilisation and cost outcomes of Yishun Zone residents by needs-based sub-segments.

	A1	A2	A3	A4	A5	B1	B2	B3	B4	B5	*p*-Value
**Average No. of ED visits**	0.6	0.4	0.9	0.9	1.6	0.7	0.7	1.3	1.1	2.1	**<0.001**
**Average No. of SOC visits**	2.0	2.7	3.0	3.5	3.7	2.6	3.4	3.5	4.9	4.8	**<0.001**
**Average No. of inpatient admission**	0.3	0.4	0.5	0.7	1.0	0.6	0.7	1.1	1.3	2.0	**<0.001**
**Average annual hospitalisation bed days**	0.7	0.8	2.3	4.8	9.5	2.8	2.7	8.1	9.9	22.6	**<0.001**
**Average annual overall healthcare cost (SGD)**	$1941	$2538	$3981	$6955	$10,682	$5432	$6018	$10,399	$14,524	$23,356	**<0.001**

**Table 4 healthcare-13-02724-t004:** Healthcare utilisation, cost, and patient outcome by episodic segments.

	ES1	ES2	ES3	ES4	ES5	ES6	*p*-Value
**Number of cases** **(%)**	1966 (11.7%)	5807 (34.6%)	4215 (25.1%)	1274 (7.6%)	722 (4.3%)	351 (2.1%)	
**Average length of stay (Day)**	2.2	4.3	8.2	6.7	11.8	10.7	<0.001
**Average inpatient admission cost (SGD)**	$3283	$5571	$8384	$14,314	$17,298	$10,655	<0.001
**30-day emergency readmission rate**	1.4%	5.3%	16.3%	6.8%	16.1%	29.9%	<0.001

## Data Availability

The data presented in this study are not publicly available due to privacy and confidentiality restrictions, as they contain patients’ medical and social information. The dataset was derived from Yishun Health’s Central Data Repository (CDR) containing administrative, clinical, and financial data. Requests to access the aggregated datasets may be considered by the corresponding author upon reasonable request and subject to approval from the relevant institutional data protection office and governance committees.

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
