# Peer review of "Systemic Population Segmentation Based on the Unified Care Model: An Approach to Health System Transformation"

_healthcare, 2025, doi:10.3390/healthcare13212724_

Round 1
Reviewer 1 Report
Comments and Suggestions for Authors
Dear Authors:
This reviewer appreciates the importance of population health in management of the wellbeing at a national level. The manuscript is well-thought out and details information that is relevant to national, state, regional, plan, and healthcare systems involved in care delivery.
Strengths of the manuscript (Review comments that do not require author comment or revisions)
- The topic inclusive of care internationally to an aging population with escalating costs and care needs is resonant to a wide global community.
- Researchers have defined carefully defined segmentation and its models to help outline the approach that defined their three interdependent models Lifelong Care Segmentation, Needs-Based Sub-Segmentation, and Episodic Care Segmentation Model.
- The population for the model description is sizable based upon 320k residents from a geographic catchment and estimated 78k included in the population cohort. The approach applied to the population is sophisticated and incorporates medical, social, and mental health with life oriented segmentation.
- Authors detail appropriate limitations of their findings.
Weaknesses of the manuscript: (Reviewer comments that do not require author comment or revisions)
- The initial description of the approach does provide depth in descriptions of the model and segmentation however, at times the content is somewhat descriptive (academic) versus operationally oriented (pragmatic) in its interpretation. While population health covers many broad areas, this reviewer feels that overall the readability would be improved if the leading 2-3 target areas addressed were threaded through the manuscript to help to contextualize for the reader. While authors aimed to describe these terms carefully in the beginning and end of the manuscript, this reviewer found them to be more difficult to follow.
- Authors share their findings in that (1) only a smaller subset of patients have high utilization, complexity, and need (2) cooccurring mental health and/or social issues resulted in significantly higher healthcare utilization and cost. These have been demonstrated in other cohorts.
Opportunities for improvement or corrections that this reviewer advises for corrections in the manuscript:
- Recommendation for further clarification of the mesosystem and microsystem: This reviewer found the description of the macro, meso, and microsystems a bit confusing early in the manuscript. While it is clear that the authors appropriately aimed to define these terms, consider a singular example in the section 1.1 to help the reader follow along very clearly with the term, definition, and example that is pulled through all of them. The authors did try examples but this reviewer thinks that taking a challenge, i.e. memory loss with aging and then walking through these from macro, to meso, to micro might help it be clearer. Either way, this reviewer recommends a slight modification in lines 63-81 to improve the readability.
- Recommendation for improvement in the readability of the sentence row 134 Salient features that proceeds until line 149. This reviewer found this paragraph to be complex and would recommend breaking up the sentence into 3 or so sentences.
- Rationale for selection of the ICD: While authors describe a list of 38 chronic illnesses and their ICD-10 codes was used to identify if (line 274) a resident had prior chronic disease diagnoses (Table 1), it was not clear why these were particularly selected.
- Authors could further expand upon the Systemic Health System Population Segmentation Model to better enable the readership community to be able to adopt or understand critical next steps in adoption of such a model.
- Authors might further expand on how their findings mirror prior findings in the population utilization and impact of psycho-social need. Their approach could have an impact that is amplified as relevant globally by other communities. To achieve this, authors may consider an additional reference and adding this to the discussion.
Recommendation for minor correction of errors or formatting that are straightforward editing corrections:
- Reformat the below sentence into two sentences on line 175 to improve the readability:
While this does not mean that the desired person-centred outcomes are not important 175
or not being attained, it does, however, mean that segmentation-specific interventions that 176
are based on population segmentation models that are not needs based or that are not 177
systemically linked to macrosystem segmentation models are unlikely to be effectively 178
designed and even less likely to be efficient, in terms of improving person-centred whole 179
health system outcomes.
- Line 284: The ‘I’ should be uncaptialized ‘i’ social Issues.
Reviewer 2 Report
Comments and Suggestions for Authors
This paper outlines the creation and testing of a systemic, person-centered, needs-based population segmentation model by Yishun Health in Singapore. It presents three interlinked models—Lifelong Care Segmentation, Needs-Based Sub-Segmentation, and Episodic Care Segmentation—within the Unified Care Model (UCM). The models strata the residents by biopsychosocial needs and navigate integrated care delivery, resource allocation, and outcome monitoring.
Areas for Improvement
- Data Limitations: Segmentation is dependent largely on administrative information, which excludes important psychosocial and functional measurements. The limitation is recognized but requires more extensive discussion of mitigation strategies.
- Model Validation: Although the expert-based approach is rational, lack of external validation or comparison with data-driven models (e.g., clustering, latent class analysis) can restrict robustness.
- Operational Feasibility: The paper would be improved by additional information regarding how these models are implemented within clinical practices, such as difficulties encountered by care teams.
- Equity Considerations: The segmentation incorporates proxies for socioeconomic status (e.g., public rental housing), yet a more direct consideration of equity and access throughout segments would enhance the analysis.
- Questions for the Authors
- Model Adaptability: How would you see the segmentation models being adapted to populations with varying demographic or epidemiological characteristics (e.g., rural vs. urban, younger populations)?
- Psychosocial Data Collection: What are the particular instruments or models being thought of to collect more robust psychosocial and functional data in subsequent versions?
- Care Team Involvement: How have front-line clinicians reacted to the segmentation models? Are there processes for their input to be used in model improvement?
- Outcome Attribution: With overlapping interventions across segments, how do you allocate improvements in outcomes to particular segmentation-driven strategies?
5.Ethical Considerations: Have you thought about possible ethical issues in segmenting populations, especially regarding prioritizing resources or stigmatization?
Recommendation
Major Revision: The manuscript presents a valuable and innovative contribution to population health system design. Addressing the questions above and expanding on implementation and validation strategies would enhance its impact and clarity.
Reviewer 3 Report
Comments and Suggestions for Authors
This is a good paper and an important contribution to the literature on health system transformation and population segmentation. The authors present a systemic, person-centred, and needs-based segmentation approach grounded in the Unified Care Model and apply it to a large resident population in Singapore. The manuscript is comprehensive, methodologically transparent, and policy-relevant. Still, there are areas where clarity, conciseness, and interpretation could be improved.
Title
-
The title is informative but overly lengthy and complex.
-
Repetition of “Systemic” and “Population Segmentation” makes it harder to read.
-
A shorter, sharper title would improve readability and impact, for example:
-
“Needs-Based Population Segmentation for Health System Transformation: A Singapore Case Study”
-
or “Systemic Population Segmentation Models for Integrated Care in Singapore.”
-
-
Streamlining the title would also help highlight the novelty (needs-based, person-centred, systemic approach) more effectively.
Abstract
-
The abstract is informative and well structured. However, it is rather dense and could be shortened for clarity.
-
The novelty of the approach (systemic, needs-based segmentation across all levels of the health system) should be emphasized more clearly.
-
Consider briefly highlighting the implications for broader international contexts, not only Singapore.
Introduction
-
The introduction provides a strong conceptual background on population segmentation, but it is quite long and occasionally repetitive.
-
The research gap could be articulated more explicitly—why existing segmentation models are insufficient, and how this study addresses these limitations.
-
The local Singapore context is clearly described but could be streamlined to avoid excessive detail.
Methods
-
The description of the Lifelong Care, Needs-Based Sub-Segmentation, and Episodic Care models is detailed and well supported.
-
However, the methodology could be clearer on:
-
How “psychosocial issues” were operationalized (e.g., mental health diagnoses, social worker involvement, housing status).
-
Why an expert-driven approach was chosen instead of a data-driven approach, and how this may affect generalizability.
-
The handling of missing or incomplete data.
-
-
Statistical analysis is appropriate, but the rationale for choosing certain tests (e.g., ANOVA vs regression) could be explained briefly.
Results
-
The results are comprehensive, with detailed tables and figures.
-
However, some tables could be simplified or moved to an appendix to improve readability.
-
The findings regarding the impact of psychosocial issues on outcomes are very important and could be highlighted more strongly.
-
The clinical and demographic characteristics (age, comorbidity index) are well presented but could be summarized more concisely.
Discussion
-
The discussion effectively situates the findings within the broader context of health system transformation.
-
However, parts of the discussion repeat the results rather than providing deeper interpretation.
-
It would strengthen the paper to discuss:
-
Comparisons with international examples of segmentation models (e.g., Kaiser Permanente, Bridges to Health).
-
Limitations of relying on administrative data only, and how future surveys may improve accuracy.
-
Possible barriers to implementing the segmentation framework at national or international levels.
-
Conclusion
-
The conclusion is clear and policy-oriented.
-
However, it could be more concise and avoid repeating discussion points.
-
Consider emphasizing the transferability of the model beyond Singapore, which would increase the international relevance of the work.
Minor Issues
-
The manuscript could benefit from careful language editing to reduce repetition and improve flow.
-
Some acronyms (e.g., NBSSM, ES) could be explained again in later sections for reader clarity.
-
The acknowledgement of AI-assisted proofreading is transparent but may be shortened.
Round 2
Reviewer 2 Report
Comments and Suggestions for Authors
The revised article can be accepetd in its current form.